# Effect of Posttranslational Modifications on the Structure and Activity of FTO Demethylase

**DOI:** 10.3390/ijms22094512

**Published:** 2021-04-26

**Authors:** Michał Marcinkowski, Tomaš Pilžys, Damian Garbicz, Jan Piwowarski, Damian Mielecki, Grzegorz Nowaczyk, Michał Taube, Maciej Gielnik, Maciej Kozak, Maria Winiewska-Szajewska, Ewa Szołajska, Janusz Dębski, Agnieszka M. Maciejewska, Kaja Przygońska, Karolina Ferenc, Elżbieta Grzesiuk, Jarosław Poznański

**Affiliations:** 1Institute of Biochemistry and Biophysics, Polish Academy of Sciences, Pawińskiego 5a, 02-106 Warsaw, Poland; mmarcinkowski@ibb.waw.pl (M.M.); tpilzys@ibb.waw.pl (T.P.); dgarbicz@ibb.waw.pl (D.G.); piwo@ibb.waw.pl (J.P.); damian@ibb.waw.pl (D.M.); mwin@ibb.waw.pl (M.W.-S.); ewasz@ibb.waw.pl (E.S.); jasio.ibb@gmail.com (J.D.); agniesza@ibb.waw.pl (A.M.M.); kaja.przygonska@gmail.com (K.P.); 2NanoBioMedical Centre, Adam Mickiewicz University, Wszechnicy Piastowskiej 3, 61-614 Poznan, Poland; nowag@amu.edu.pl; 3Department of Macromolecular Physics, Faculty of Physics, Adam Mickiewicz University, Uniwersytetu Poznanskiego 2, 61-614 Poznan, Poland; mtaube@amu.edu.pl (M.T.); maciejgielnik@amu.edu.pl (M.G.); mkozak@amu.edu.pl (M.K.); 4National Synchrotron Radiation Centre SOLARIS, Jagiellonian University, Czerwone Maki 98, 30-392 Kraków, Poland; 5Veterinary Research Centre, Department of Large Animal Diseases and Clinic, Institute of Veterinary Medicine, Warsaw University of Life Sciences, Nowoursynowska 100, 02-797 Warsaw, Poland; karolina_ferenc@o2.pl

**Keywords:** ^EC^FTO, ^BES^FTO, phosphorylation, calcium, dimerization, nanoDSF, MST, HDX, SAXS

## Abstract

The FTO protein is involved in a wide range of physiological processes, including adipogenesis and osteogenesis. This two-domain protein belongs to the AlkB family of 2-oxoglutarate (2-OG)- and Fe(II)-dependent dioxygenases, displaying *N*^6^-methyladenosine (*N*^6^-meA) demethylase activity. The aim of the study was to characterize the relationships between the structure and activity of FTO. The effect of cofactors (Fe^2+^/Mn^2+^ and 2-OG), Ca^2+^ that do not bind at the catalytic site, and protein concentration on FTO properties expressed in either *E. coli* (^EC^FTO) or baculovirus (^BES^FTO) system were determined using biophysical methods (DSF, MST, SAXS) and biochemical techniques (size-exclusion chromatography, enzymatic assay). We found that ^BES^FTO carries three phosphoserines (S184, S256, S260), while there were no such modifications in ^EC^FTO. The S256D mutation mimicking the S256 phosphorylation moderately decreased FTO catalytic activity. In the presence of Ca^2+^, a slight stabilization of the FTO structure was observed, accompanied by a decrease in catalytic activity. Size exclusion chromatography and MST data confirmed the ability of FTO from both expression systems to form homodimers. The MST-determined dissociation constant of the FTO homodimer was consistent with their in vivo formation in human cells. Finally, a low-resolution structure of the FTO homodimer was built based on SAXS data.

## 1. Introduction

A strong association between Single Nucleotide Polymorphisms (SNPs) in the first intron of the human *FTO* (FaT mass and Obesity-associated) [1] gene and an increased body mass index (BMI) was reported in 2007 [2,3]. The discovery prompted extensive studies on the biological function of the *FTO*-encoded protein [4]. Later data suggested that the mentioned SNPs in question did not in fact affect the *FTO* gene but instead changed the regulatory sequence of another obesity-associated gene located on chromosome 16, *IRX3* [5]. Nevertheless, the interest in FTO protein has led to the discovery of its engagement in a wide range of key physiological processes such as adipogenesis [6], osteogenesis [7], neural development [8], heart remodeling [9] and cell cycle progression [10], as summarized in Figure 1A. Moreover, FTO dysfunctions lead to major disturbances like cancer [11,12,13,14,15,16] and improper brain development [17]. These multifarious functions of FTO likely reflect its different substrate specificities in different cell types. FTO is a demethylase and so far has been shown to remove the methyl group from the following modified nucleosides in single-stranded DNA or RNA: *N*^3^-methylthymidine [18], *N*^3^-methyluridine [19] (Figure 1B), *N*^6^-methyladenosine (*N*^6^-meA), *N*^1^-methyladenosine [20] and, the most efficiently, from *N*^6^,2′-*O*-dimethyladenosine (*N*^6^-meA_m_) in the mRNA cap [21] (Figure 1C).

The catalytic center of FTO is located in the N-terminal dioxygenase domain (residues 1–327) where the methyl group is removed in the presence of O_2_ and Fe^2+^ as cofactors and 2-oxoglutarate (2-OG) as a co-substrate (Figure 1B) [18]. The positioning of the C-terminal domain (residues 328–505; Figure 1D) allows the catalytic domain to maintain proper structure and thus an ability to remove the methyl group from *N*^3^-methylthymidine [22]. Summing up, FTO removes chemical modifications such as *N*^6^meA and *N*^6^meA_m_ from various RNA classes (mRNA, tRNA, snRNA) affecting RNA stability, efficiency of translation, degradation rate or even alternative splicing [20,23].

Despite the plentitude of studies on the involvement of FTO in metabolic regulation, relatively few data concerning the relationship between FTO structure and function have been published. Several phosphorylated amino acids have been identified in FTO [24], and their phosphorylation status shown to affect the protein fate. Tai and co-workers reported that threonine phosphorylation by protein kinase Cβ increased the FTO lifetime [25], while Faulds and co-workers showed that in mice phosphorylation of specific serines in the N-terminal domain—S249 and S253—led to FTO degradation [26]. A recent study by Hirayama and co-workers has indicated that phosphorylation of a specific threonine, T150, affects the cellular localization of FTO and its ability to remove N^6^meA from particular transcripts [10], thereby modulating the FTO substrate specificity [20]. However, only the latter study directly addressed the influence of FTO phosphorylation on its activity, and only one phosphorylation site was studied, indicating that this field is far from being thoroughly investigated. Additionally, the role of the C-terminal domain is poorly explored. It was pointed out to be involved in FTO homodimerization [22]. A recent paper showed that FTO functioning is modulated by the interaction of its C-terminal domain with the SFPQ protein, which allows the demethylation of specific transcripts recognized by SFPQ [27]. The recent avalanche of papers indicating the importance of various types of methylation modifications of RNA [28] combined with the well-documented engagement of FTO in key physiological processes make detailed understanding of the mode of FTO functioning and its regulation of considerable interest.

Here, we present a comprehensive analysis of the relationships between the structure and activity of FTO and of the effects of diverse modulating factors. Using biophysical methods, we determined the influences of the FTO phosphorylation status, small molecules and ions on its properties. To determine the effects of post-translational modifications we compared the characteristics of recombinant N-terminal His-tagged human FTO obtained from the prokaryotic *Escherichia coli* (EC) expression system with those from the eukaryotic baculovirus expression system (BES), in which posttranslational modifications mimicking those in the native mammalian FTO could be expected.

## 2. Results

### 2.1. Thermal Stability of FTO Depends on Expression System Used

Unmodified FTO was expressed in *E. coli* (^EC^FTO), whereas the protein modified in a manner likely mimicking that of the native mammalian protein was obtained in the eukaryotic baculovirus expression system (^BES^FTO). Highly pure FTO preparations were obtained from both the systems (Figure 2A), and both were capable of *N*^6^-meA demethylation with similar efficiency (Figure 2B, lanes 1 and 2, respectively).

A spontaneous mutation was found in ^BES^FTO causing the S246G replacement, however in silico calculations showed that this change only slightly affected the protein stability (~1.2 kcal/mol, Appendix A). We also confirmed experimentally that the enzymatic activity of ^EC^FTO^S246G^ is close to that of the wild type ^EC^FTO (Appendix A, lanes 1 and 2, respectively).

Subsequently, we analyzed thermal unfolding of apo (without cofactors) and holo (with 2-OG and Mn^2+^—non-catalytic cation mimicking Fe^2+^) forms of both FTO preparations using low-volume Differential Scanning Fluorimetry (nanoDSF) (Figure 2C). The thermal denaturation curves displayed two peaks indicative of two transitions likely representing sequential unfolding of the N- and C-terminal domains, which includes residues 1–327 and 328–505, respectively. The thermal stability of the apo form was almost identical for both preparations. The holo form was markedly more stable, and there was a statistically significant difference in the first unfolding temperature, 72 ± 1 °C and 65 ± 3 °C between ^EC^FTO and ^BES^FTO, respectively (*p* < 0.05), and a smaller one for the second transition. Taken together, these data indicate that the structural properties of FTO differ slightly between the two expression systems used.

Since the holo form differed markedly from the apo one, we monitored effects of the two cofactors, 2-OG and Mn^2+^, used separately. Both of them increased the T_m_ significantly, indicating an interaction with FTO (Appendix A). In the presence of Mn^2+^ the first transition was shifted towards higher temperatures, while the second one remained unaffected. An opposite effect was observed for 2-OG which affected solely the high-temperature transition. Interestingly, in the holo form, with the both ligands, the low-temperature transition occurred at an even higher temperature than in the presence of Mn^2+^ alone, while the high-temperature transition was identical as that in the presence of 2-OG alone. All these observations indicate that the more affected low-temperature transition corresponds to the thermal denaturation of the N-terminal domain, the catalytically active state of which harbors both cofactors. However, it has to be noted that nanoDSF follows the effect of ligand binding at non-physiological temperatures, at which the protein unfolds. To avoid this problem, we used Microscale Thermophoresis (MST) to study Fe^2+^ and 2-OG binding at the biologically relevant temperature of 25 °C. 2-OG affected the Fe^2+^ binding by FTO marginally (*K*_D_ = 6.0 ± 2.4 and 7.2 ± 2.1 µM in the presence and absence of 2-OG, respectively, with the consensus *K*_D_ value of 7.0 ± 2.4 µM estimated using global fitting; see Figure 2C), while 2-OG bound to FTO only in the presence of Fe^2+^ (*K*_D_ = 5.2 ± 0.8 µM in the presence of Fe^2+^ and 800 ± 200 µM in its absence; see Figure 2D).

### 2.2. Posttranslational Modifications Affect FTO Demethylating Activity

Expression of proteins in the baculovirus system allows diverse post-translational modifications to occur, including phosphorylation, glycosylation and *N*-myristoylation, similar to those observed in mammalian cells. Numerous FTO residues potentially recognized by enzymes that introduce post-translational modifications were identified using the ProSite server [29]. Among them there were 11 putative phosphorylations (T6, T32, S56, T105, T128, S256, S260, T320, S355, T393 and T443), four *N*-myristoylations (G100, G180, G182 and G412), and two *N*-glycosylations (N200 and N302) sites. Even more potential phosphorylation sites were identified in the PhosphoSitePlus [30] database that integrates both low- and high-throughput data sources. Among them was S256 as the highest occurrence, followed by T4, S260, T6, S184, S229, S173, Y199, T32, S55, Y106, Y108, T150, Y185, Y220, S355 and S458 [24]. The majority of these sites are located in the N-terminal catalytic subunit of FTO.

Following the in silico analyses, the actual phosphorylation pattern of FTO was determined experimentally by Mass Spectrometry. Three highly abundant phosphoserine residues were identified in ^BES^FTO, namely S184, S256, and S260, while no such modifications were identified in ^EC^FTO (Appendix A). Interestingly, these three serine residues are located in two regions of the N-terminal domain that are disordered in all FTO structures accessible in Protein Data Bank. Molecular modeling analyses showed that phosphorylation of S184, located in the N-terminal part of a large disordered loop covering residues 164–188, would enhance its interactions with the proximal basic residues (K162 and R178), thereby stabilizing a particular conformation (Figure 3). Moreover, the in silico analysis performed with the RaptorX server (http://raptorx.uchicago.edu; accessed on 3 April 2021) [31] showed that the partially disordered loop (N164-G187), which is invisible in all structures from Protein Data Bank, become significantly stabilized upon S184D/E replacement mimicking serine phosphorylation (Appendix A). It can therefore be expected that the S184 phosphorylation locally stabilizes the protein structure. The S256D/E and/or S260D/E replacements in in silico modelling did not affect the organization of the disordered loop ^250^EGPEEESEDDSHLEGRDPDI^269^.

We also experimentally studied the effect of phosphorylation on the stability of FTO using a thermal shift assay. ^BES^FTO was dephosphorylated enzymatically in the required buffer and, to avoid additional manipulations that would affect protein samples, nanoDSF measurements were then performed in the same buffer. In these conditions in the unfolding curve only a single transition was distinguishable rather than the two identified previously (Appendix A). The dephosphorylation of the holo form of ^BES^FTO resulted in a small, but statistically significant, decrease in the denaturation temperature (69.4 ± 0.4 °C vs. 68.0 ± 0.2 °C, *p* < 0.05), while no such effect was observed for ^EC^FTO (Appendix A). Since the both FTO preparations treated with phosphatase had the same denaturation temperature (~68.0 °C), one may deduce that ^EC^FTO and ^BES^FTO differ exclusively by their phosphorylation state.

Unfortunately, the effect of dephosphorylation of ^BES^FTO on its demethylase activity could not be studied directly since incubation in the dephosphorylation buffer, even in the absence of phosphatase, significantly decreased the enzymatic activity of both ^EC^FTO and ^BES^FTO samples by 25% and 80%, respectively (Appendix A).

To overcome this problem we introduced the S256D substitution to ^EC^FTO to mimic the most abundant phosphorylation. This resulted in a statistically significant decrease in the catalytic activity (by approximately 40%, *p*-value < 0.05; Figure 1B). Therefore, it can be expected that the phosphorylation of S256 (and possibly of S260 as well) should reduce the enzymatic activity of FTO. Notably, a direct comparison of the specific activities of ^EC^FTO and ^BES^FTO did not reveal the expected difference, most likely because of the low level of phosphorylation of ^BES^FTO. Based on MS data, the extent of phosphorylation was estimated at 3% (for S184), 37% (for S256) and 19% (for S260).

Altogether, these results show that FTO expressed in the baculovirus system, and presumably also the native mammalian FTO, can be phosphorylated at three serine residues. These modifications affect the protein stability, and at least phosphorylation of S256 leads to a reduction in the demethylase activity.

### 2.3. Calcium Affects FTO Stability

Two of the three phosphorylable serine residues (S256 and S260) lie within an extremely acidic stretch located in a solvent-exposed loop comprising residues 249–312. Including the phosphoserines, there are 12 acidic residues within a 19-amino acid-long sequence. This loop was pointed out as a putative Ca^2+^ binding site(s), and this prediction was verified experimentally.

The FTO–Ca^2+^ interaction was studied using Differential Scanning Fluorimetry (DSF) with a SYPRO^®^ ORANGE probe (Figure 4A). Both the apo and holo forms of ^BES^FTO showed slightly (by 0.8 and 0.7 °C, respectively) but significantly (*p* < 0.05) increased transition temperatures in the presence of Ca^2+^ (Figure 4A). The effect of calcium on the thermal stability of FTO remains independent of the presence of Fe^2+^ (holo vs. apo), suggesting that there is no direct interaction between Ca^2+^- and Fe^2+^-binding sites. We used further hydrogen–deuterium exchange monitored with MS (HDX-MS) to identify the region(s) affected by Ca^2+^ binding. Two regions stabilized in the presence of Ca^2+^ were found, comprising residues 120–155 and 240–310, respectively (Figure 4B). Interestingly, they are close to each other in the protein structure (Figure 4C). The second region encompasses the acidic stretch comprising S256 and S260—the two phosphorylable serines mentioned above. This confirms the prediction that the 249–312 loop is involved in Ca^2+^ binding. However, the stabilizing effect of Ca^2+^ is rather small, in agreement with the surface location of this loop. Notably, H307, which is also located in the second region stabilized by calcium presence, is involved in the coordination of the Fe^2+^ ion required for the demethylase activity. Therefore, it can be hypothesized that Ca^2+^ is likely to modulate the enzymatic activity of FTO, by indirect interaction with the residue that is involved in Fe^2+^ binding. This prediction was verified experimentally.

### 2.4. Ca^2+^ Affects FTO Catalytic Activity

Although the stabilizing effect of Ca^2+^ was rather small, it affected regions of the N-terminal domain of ^BES^FTO involved in enzymatic activity. Indeed, Ca^2+^ was found to decrease the activity of both FTO preparations (Figure 5A,B). Despite its marginal effect on FTO stability, Ca^2+^ decreased the demethylase activity significantly, in the case of ^BES^FTO by more than 80%. Moreover, the stronger effect observed for ^BES^FTO agreed with our in silico predictions and HDX-MS data that phosphoserines 256 and 260 may contribute to Ca^2+^ binding.

### 2.5. FTO Forms Homodimer via Its C-Terminal Domain

We also investigated how the phosphorylation affects the FTO protein–protein interaction by examining the ability of ^EC^FTO and ^BES^FTO to form homodimers in the presence of Mn^2+^ and 2-OG. Both cofactors were present in experimental solution, and in elution buffer as well, to be positive that holo state of the FTO was under investigation. Upon gel filtration of protein samples of approximately 5 µM concentration, both preparations eluted as asymmetric peak at dominating fraction roughly corresponding to mass ~80 kDa with a shoulder at ~50 kDa (molecular mass of FTO is 58 kDa) (Figure 6). Such a shape of the elution curve is indicative for the dynamic exchange between the homodimeric and monomeric forms of the protein [32], which results in apparently underestimated mass of the FTO homodimer (expected 116 kDa). On the other hand, the underestimated mass of the FTO monomer agrees with the non-spherical elongated shape of the protein (see Figure 7).

The organization of the FTO homodimer was studied using SAXS (Figure 7A,B). A low-resolution model fitted to the scattering data (approximate resolution 12–20 Å) supported the presence of a homodimer (Figure 7A)—the experimental gyration radius was approximately 38 Å, while the value for a monomer estimated with the Yasara Structure was 27 Å. The mass (84.8 kDa) estimated from the SAXS data (Figure 7A) also exceeded the mass of the monomer (58 kDa) and was consistent with the gel filtration results. A further structural refinement was performed by analyzing 100 models of the ^BES^FTO homodimer generated with the Symmdock server. Their theoretical scattering curves agreed with the experimental data (Figure 7B), for two of those models only (Figure 7C,D). In one model, the dimerization occurs via the C-terminal domain, while in the other it occurs through the N-terminal domain. However, only for model 80 does the gyration radius (37.8 Å; Figure 7C) agree with the experimental value of 37.8 ± 0.5 Å.

We used the proton–deuter exchange data to further evaluate the two models. HDX-MS identified the two most protected regions of ^BSE^FTO—residues 130–160 and 330–420 (Figure 8A). In the monomer, the first one is a buried alpha-helix located in the N-terminal domain and the second one is a group of surface-exposed residues of the C-terminal domain. Largely reduced H/D exchange in the latter region agrees with it being the interface for FTO dimerization, as shown in Figure 7C. Thus, both the SAXS and HDX-MS data consistently indicate that the C-terminal domain is responsible for the FTO dimerization. The predicted structure of an FTO dimer is presented in Figure 8B.

The stability of FTO dimers (*K*_D_) was estimated using MST. We titrated fluorescently labelled FTO with an unlabeled one, in the absence or presence of Fe^2+^ and/or 2-OG. In the presence of both cofactors the *K*_D_ for the ^BES^FTO dimer was 1.99 ± 0.26 µM. ^EC^FTO displayed a *K*_D_ value (1.79 ± 0.20 µM) close to that determined for ^BES^FTO (Figure 9). Unexpectedly, with only one of the cofactors, either Fe^2+^ or 2-OG, or in their absence the ^BES^FTO dimers were more stable (*K*_D_ = 0.34 ± 0.07 µM, see Figure 6G–I for details). This suggests that the binding of cofactors affects not only the active site of FTO located in the N-terminal domain, but also disturbs the dimerization interface involving C-terminal domains. The *K*_D_ values of holo form of ^EC^FTO and ^BES^FTO are consistent with the results of the gel filtration experiment, in which 5 µM protein samples were analyzed. At this concentration, the dimeric form of the holo protein dominates (approximately 70%), but the monomeric form is still fairly abundant. A *K*_D_ value in the micromolar range does not preclude in vivo existence of FTO dimers.

## 3. Discussion

The FTO protein, a member of the ALKBH family, has been found to be associated with obesity and other diseases of affluence like type 2 diabetes and cancer. Molecular studies have revealed that it is a demethylase engaged in the reversion of the *N*^6^-meA modification in mRNA and thereby in the regulation of metabolism at the posttranscriptional level [33]. Taking into consideration how many crucial processes are affected by FTO, surprisingly little is known about the regulation of its activity, its exact substrate specificity, and the relationship between the FTO structure, activity, and interactome. Here, we performed a detailed biophysical and biochemical characterization of purified FTO samples expressed in two heterologous systems, in the presence or absence of its known or putative cofactors. By comparing FTO expressed in a bacterial and a eukaryotic expression system we were able to determine the effects of the most common eukaryotic protein modification—phosphorylation. We took advantage of the fact that BES is known to mimic the protein phosphorylation in mammalian cells [34]. In line with that assumption, we identified three phosphoserine residues in ^BES^FTO and none in ^EC^FTO. All three modified residues lie in solvent-exposed loops; two of them, S256 and S260, have been reported earlier [26] while S184, in the N-terminal domain of FTO, is novel.

No significant differences in the enzymatic activity were observed between ^EC^FTO and ^BES^FTO. However, the extent of phosphorylation of the three serines was well below 100% and in fact the ^BES^FTO preparation was likely a mixture of forms carrying between zero (this form, according to MS data, was dominant) and three phosphoserines. If the effect of phosphoserines on enzymatic activity was small, it could well be within the experimental error in such a heterogeneous population. On the other hand, that partial phosphorylation was sufficient to markedly decrease the overall stability of FTO, as evidenced by the different temperatures of the first thermal transition of ^BES^FTO and ^EC^FTO. Additionally, the loss of enzymatic activity after incubation under dephosphorylation conditions was higher for ^BES^FTO. These differences between the two FTO preparations were observed for the holo form of the protein. Taken together these results indicate that phosphorylation may affect FTO properties to some extent.

It is also possible that phosphorylation affects FTO functioning indirectly, e.g., by modulating its intracellular localization [10], interactions with other proteins, or susceptibility to degradation [26]. One cannot also exclude a selective influence of FTO phosphorylation(s) on its demethylase activity towards certain substrate only [20]. All these possibilities would not be picked up by a simple in vitro activity test missing the cellular context.

To determine whether the apparent lack of a difference in activity between ^BES^FTO and ^EC^FTO stems from the incomplete phosphorylation of the former or latter reflects the context-dependent effect of phosphorylation, as speculated above, we introduced a phosphomimetic point mutation S256D into ^EC^FTO. This approach allowed a convincing demonstration that the mutant exhibited significantly lower activity than the wild-type ^EC^FTO. It is worth mentioning that murine phosphorylation on serines corresponding to S256 of human FTO is the first step of a cascade leading to proteolytic degradation of FTO [26]. These facts combined suggest a sequence of events with phosphorylation initially decreasing FTO activity followed by its degradation by the cellular machinery.

Inspection of the amino acid sequence in the vicinity of the two nearby phosphorylated serine residues (S256 and S260) in the N-terminal uncovered a putative calcium-binding site in an unstructured region on the FTO surface, absent from its crystallographic structure (pdb4IE5), with a highly acidic loop (residues 250-260) potentially involved in Ca^2+^ binding [35,36]. The S256 and/or S260 phosphorylation could, theoretically, enhance this binding even further. Ca^2+^ had a small stabilizing effect on FTO, as determined using the DSF and HDX, much smaller than that exerted by the two cofactors, Fe^2+^ and 2-OG, bound in the catalytic center. This was in line with the proposed peripheral location of the calcium-binding site as both the techniques detect mostly changes in the hydrophobic core of the protein (DSF) or in secondary structures (HDX) [37,38]. In contrary to our prediction, the structural stabilization by Ca^2+^ was similar for ^EC^FTO and ^BES^FTO arguing against a major role of phosphoserines in the calcium binding.

Notably, calcium signaling may also regulate ubiquitination [39]. As mentioned earlier, phosphorylation of FTO in mice led to ubiquitination [26] suggesting that the effects of Ca^2+^ on FTO deserve further investigation.

We also found that Ca^2+^ inhibited the enzymatic activity of both ^EC^FTO and ^BES^FTO. Numerous metabolic processes are known to be regulated by calcium, often via [40] the ubiquitous Ca^2+^-binding protein calmodulin (CaM), a promiscuous interactor [41].

While characterizing the biophysical properties of FTO, we also sought to solve the question of its oligomerization. Earlier studies indicated that FTO can form complexes of higher molecular mass. Basing on non-denaturing electrospray ionization mass spectrometry and analytical gel filtration, Church and coworkers proposed that FTO forms a homodimer through its C-terminal domain [22]. On the other hand, Han and coworkers obtained a crystallographic structure with FTO in a trimeric form, which existence was however assigned to the crystal packing [18]. Moreover, a tendency of dioxygenases to form dimers was shown earlier [42,43]. Here, using several independent methods (gel filtration, MST, SAXS), we showed that in solution FTO tends to dimerize, with a dimer dissociation constant in the micromolar range. Such concentrations of FTO may occur in specific cellular compartment like the nucleus. Modeling showed that the dimerization most likely occurs via the C-terminal domains, as proposed earlier [22]. Notably, phosphorylation did not affect the dimer stability significantly.

Interestingly, the *K*_D_ of the ^BES^FTO dimer is lower for the apo form than for its catalytically active holo form. However, determination of the exact impact of the dimerization on the FTO catalytic activity is not easy, due to the direct interaction between the N- and C- domains [22]. When co-expressed as separate polypeptides in *E. coli*, these domains formed a stable complex [18]. Additionally, computational studies of Waheed and coworkers have revealed that the N- and C-terminal domains move with respect to one another in a manner likely important for substrate binding [44]. Therefore, testing the influence of the dimerization on activity by mutating amino acids in the C-terminal domain would be questionable. Such mutations could affect the activity, not by blocking the dimerization, but by changing the contact surface between the N- and C-terminal domains. Additionally, Han and coworkers made the same argument [18], when commenting on Church and coworkers’ article [22]. If, indeed, the FTO dimer is less active than the monomer, then at low FTO concentrations the dimerization could help keep its activity constant despite fluctuations in the local protein abundance. Interestingly, FTO level is highly elevated in head and neck cancer and correlates with tumor development [16]. This implicates its specific role in cancer cells and adds to the interest in FTO regulation.

## 4. Materials and Methods

### 4.1. cDNA Cloning and Plasmid Constructs

Plasmid pET-28a(+) harboring cDNA encoding full length human FTO His-tagged at the N-terminus was kindly provided by prof. Arne Klungland (Oslo University Hospital, Oslo, Norway).

For use in the Baculovirus Expression System (BES), the coding sequence was PCR-amplified with suitable primers (Appendix A) and inserted into the pENTRY-IBA5 donor vector between the Xbal and HindIII sites. The insert was then moved to the pLSG-IBA35 destination vector (StarGate cloning system, IBA Life Sciences, Goettingen, Germany).

### 4.2. Site Directed Mutagenesis

Site directed mutagenesis (SDM) was performed in 50 µL of SDM1 (composition of all buffers is presented in Table 1) buffer. Mutated plasmids were amplified in an Eppendorf Mastercycler^®^ according to the program: 94 °C for 2 min, followed by 18 cycles of 94 °C for 20 s, 55 °C for 30 s, 68 °C for 7 min and final 68 °C for 10 min. The primers used for SDM are shown in Appendix A. Non-mutated plasmids were removed by incubation with DpnI: 50 µL of sample was mixed with 6 µL of 10x Tango buffer and 4 µL of DpnI (10U/µL, Thermo Scientific™, Waltham, Massachusetts, USA, #ER1701), and incubated with mixing (500 rpm) for 2 h at 37 °C. The reaction was stopped by incubation at 80 °C for 20 min. Then, 8 µL of reaction mixture was added to 200 µL of competent *E.coli* DH5α bacteria suspension, incubated for 30 min on ice, heat shocked for 30 s at 42 °C, then 800 µL of SOC buffer was added and the bacterial suspension was incubated at 37 °C for 1 h. The bacteria were plated on LB-agar plates and incubated overnight at 37 °C. Single colonies were cultured in liquid LB overnight at 37 °C. Bacteria were then harvested and the plasmid was isolated with the use of the GeneJET Plasmid Miniprep Kit (Thermo Scientific™, Waltham, MA, USA, #K0503). All plasmids were verified by sequencing.

### 4.3. Expression and Purification of FTO in E. coli

Plasmids harboring cDNA encoding N-terminally His-tagged full length human FTO, or its mutated variants S246G or S256D, were introduced into *E. coli* BL21. Bacteria were cultured at 37°C to OD_600_ = 1 in 2 L of LB supplemented with kanamycin (50 µg/mL) and chloramphenicol (25 µg/mL), induced with 1 mM IPTG, cultured for 16 h at 16°C, harvested, resuspended in PRP1 lysis buffer and incubated at room temperature for 30 min. After centrifugation (20,000× *g*, 10 min), the His-tagged proteins were purified in three steps. First, overnight incubation of the sample with 800 µL of Ni-charged resin suspension (Profinity™ IMAC Ni-Charged Resin, BIO-RAD, Hercules, CA, USA, #156-0133) was performed at 4 °C. The unbound fraction was removed by washing three times with lysis buffer. Next, elution was performed with PRP2 buffer; the released, protein was subjected to size-exclusion chromatography on a SEC650 (ENrich™ SEC 650, BIO-RAD, Hercules, CA, USA, #780-1650) column, and finally dialyzed overnight into PRP3 buffer. Protein purity was verified by SDS-PAGE. The purified protein was flash-frozen in liquid nitrogen and stored at −80 °C.

### 4.4. Expression and Purification of Proteins in Baculovirus Expression System (BES)

The recombinant baculoviruses were generated directly in Sf21 insect cells by co-transfection with plasmid containing the human *FTO* coding sequence and FlashBacUltra virus DNA following the manufacturer’s protocol (FlashBAC™ system, Oxford Expression Technologies, Oxford, UK). Insect cells were infected with a recombinant baculovirus at a MOI = 4. After 72 h the cells were harvested, resuspended in PRP4 lysis buffer and homogenized. After centrifugation (20,000× *g*, 10 min), proteins His-tagged at the N-terminus, were purified, verified by size exclusion and SDS-PAGE, and stored as described in the previous section.

### 4.5. Preparation of Protein Samples

Purified protein was thawed on ice, centrifuged (30,000× *g*, 30 min), the supernatant was supplemented with EDTA (pH 8.0) to 0.5 mM and incubated on ice for 10 min. FTO concentration was determined by measuring absorbance at 280 nm (assuming ε = 96215 M^−1^cm^−1^ for ^EC^FTO, ^EC^FTO^S246G^, ^EC^FTO^S256D^ and ^BES^FTO). The proteins were transferred into a suitable target buffer with the use of salting-out columns (Zeba Spin Desalting Column 7K MWCO, Thermo Scientific™, Waltham, MA, USA, #89882).

### 4.6. Molecular Modeling

Since in all FTO structures deposited in Protein Data Bank the loops ^166^DAVPLCMSADFPRVGMGSSYNG^187^ and ^251^GPEEESEDDSH^261^ are disordered, their coordinates were reconstructed using the BuildLoop algorithm [45] implemented in the Yasara Structure package (ver. 19.5.5; https://www.yasara.com; accessed on 1 February 2021). The thermodynamic effect of substitutions of S184, S246, Ser256 and S260 on the stability of the FTO protein was assessed using FoldX software [46]. The latter calculations included the contributions of the S246G replacement and experimentally identified phosphorylations of S184, S256 and S260.

### 4.7. Enzymatic Assay

FTO activity was determined using the FTO Chemiluminescent Assay Kit (BPS Bioscience). Before reaction, FTO samples purified from either *E. coli* or BES were incubated with 0.5 mM EDTA for 5 min and then underwent one of two procedures: buffer exchange on a salting-out column (Zeba™ Spin Desalting Columns, 7K MWCO, 0.5 mL, Thermo Scientific™, Waltham, MA, USA, #89882) or overnight dialysis. The measurements were carried out according to the manufacturer’s protocol with RNA containing *N*^6^-meA (provided in the kit) as a substrate. Chemiluminescence was measured using the Synergy HT (Bio Tek, Winooski, USA) plate reader with an integration time of 1 s and 0.1 s delay after the plate movement. The demethylase activity of a given FTO sample was estimated according to the following formula, and further adjusted for protein concentration.
Relative activity = (S_sample_ − S_bgk_)/(S_ECFTO_ − S_bkg_) × 100%(1)
where S_sample_ and S_bkg_ are the chemiluminescence intensities for reaction containing a protein sample and the buffer background, respectively, and S_ECFTO_ is the mean signal intensity by a reference sample containing 1 µg of purified ^EC^FTO and determined for 2–3 independent preparations.

### 4.8. Screening for Ligand Binding

Thermal stability of the protein was followed by Differential Scanning Fluorimetry (DSF; 4.8.1) or low-volume Differential Scanning Fluorimetry (nanoDSF; 4.8.2), in the presence of various combinations of ligands/cofactors: (NH_4_)_2_Fe(SO_4_)_2,_ MnCl_2_, 2-OG, CaCl_2_. Mn^2+^ is commonly used as an inactive iron-mimicking divalent metal in the structural studies on Fe-dependent dioxygenases belonging to AlkB family [47,48,49]. According to PDB data, Mn^2+^ binds exclusively in the catalytic center of the FTO protein (4qkn, 4zs2, 4zs3, 5zmd), while the possible Mn^2+^ binding to the N-terminal His-tag [50] should not interfere with the protein stability. Importantly, Mn^2+^ does not experience spontaneous oxidation under the experimental conditions used, like Fe^2+^ undergoes in the absence of the ascorbic acid. We used Mn^2+^ solely in experiments based on UV detection, in which the presence of ascorbic acid interferes with UV detection (i.e., label-free nanoDSF and exclusion chromatography).

#### 4.8.1. Differential Scanning Fluorimetry

DSF screening based on SYPRO^®^ ORANGE emission was monitored at 580 nm (excitation at 465 nm) in the temperature range of 20–99 °C (0.57 °C/min ramp). Experiments were performed on a LightCycler480 device (Roche, Penzberg, Upper Bavaria, Germany) equipped with 384-well plates. Protein stocks were diluted to a final concentration of 1 µM in TSA1 or TSA3 experimental buffer alone or supplemented with 0.5 mM CaCl_2_ (TSA2 or TSA4, respectively). The same protein solution in the presence of 0.5 mM (NH_4_)_2_Fe(SO_4_)_2_ and 1 mM 2-OG was always used as a reference.

#### 4.8.2. Label-Free nanoDSF

Label-free nanoDSF experiments were performed on a Prometheus NT.48 device (NanoTemper Technologies GmbH, München, Germany) equipped with a holder enabling parallel measurement of 48 samples. The ratio of the natural fluorescence of the protein at 330 and 350 nm (excitation at 280 nm) was monitored at the temperature range of 20–95 °C (1 °C/min ramp). Initial protein stocks were diluted to the final concentration of 2.5 µM in DSF1 experimental buffer alone or supplemented with 0.5 MnCl_2_ (DSF2) or 1 mM 2-OG (DSF3) or both of them (DSF4). FTO samples subjected to dephosphorylation were diluted to a concentration of 3.4 µM in DSF5 experimental buffer alone as a control or supplemented with 800 U of Lambda Protein Phosphatase (DSF6). The same protein solution in the presence of 0.5 mM MnCl_2_ and 1 mM 2-OG was always used as a reference.

All measurements were repeated 2–4 times, depending on the signal-to-noise ratio. The experimental data were analyzed with R 3.3.3 using the *sm.spline* function [51]. The location of the maximum of the first derivative of the signal was interpreted as the middle-point temperature of thermal unfolding (T_m_).

### 4.9. Microscale Thermophoresis (MST)

Proteins were labeled with Nanotemper^®^ dye NT-647-NHS (lysine labeling) following manufacturer’s protocol, and their fluorescence was used to monitor the thermophoretic effect. For each labeling 1 nanomole of protein (50 µL of 20 µM solution) and 2.5 nanomoles of the dye were used. The labeling efficiency was determined according to manufacturer’s protocol. Only samples with the labeling efficiency in the range of 65–90% were used in further experiments to make sure that in most cases only one molecule of the dye would be attached to the protein. After labeling, the protein was transferred to experimental buffer MST1 alone or supplemented with 0.5 mM (NH_4_)_2_Fe(SO_4_)_2_ (MST2), 1 mM 2-OG (MST3) or both of them (MST4). Pseudo-titration experiments were performed on a Monolith NT.115 RED/BLUE device (NanoTemper Technologies GmbH, München, Germany) equipped with a holder allowing parallel measurement of 16 samples placed in capillaries. Emission of the dye was monitored at 670 nm (excitation at 650 nm by LED laser) at room temperature. The LED laser power and MST power were optimized for each system individually, according to the labeling efficiency. When ligand binding caused change of the fluorescence, the “steady fluorescence mode” was used instead of the thermophoretic effect, accordingly to manufacturer’s protocol.

It must be noted that, during labeling with the fluorescence probe, the predominance of the dimeric form of FTO protected the lysine residues at the dimer interface (putatively K126, K408 and K409) from modification, so K_D_ values determined for the labelled protein samples can be considered unperturbed by the labelling procedure.

The data were analyzed using R 3.3.3 software [52] according to a standard two-state model describing 1:1 equilibrium between the unbound and bound forms of the labeled protein [53].

### 4.10. In Silico Analysis of FTO Amino Acid Sequence

The amino acid sequence of human FTO was analyzed to identify possible regulatory elements affecting FTO functioning with the set of software: ProSite (https://prosite.expasy.org/; accessed on 1 February 2021), PhosphoSitePlus (https://www.phosphosite.org/; accessed on 2 February 2021), MyHits (https://www.expasy.org/resources/myhits; accessed on 3 February 2021), Protein–Ligand Interaction Profiler (https://projects.biotec.tu-dresden.de/plip-web/plip; accessed on 4 February 2021), COFACTOR (https://zhanglab.ccmb.med.umich.edu/COFACTOR/; accessed on 5 February 2021) and IonCom (https://zhanglab.ccmb.med.umich.edu/IonCom/; accessed on 8 February 2021).

### 4.11. Identification of FTO Phosphorylation Sites

Protein samples were subjected to standard in-solution digestion. Purified protein was reduced with 200 mM dithiothreitol (30 min at 60 °C), alkylated with 500 mM iodoacetamide (15 min at room temperature in darkness) and digested overnight with trypsin (Sequencing Grade Modified Trypsin—Promega V5111). The resulting peptide mixture was analyzed by LC-MS-MS/MS (liquid chromatography coupled with tandem mass spectrometry) using a Nano-Acquity LC system (Waters Corporation, Milford, MA, USA) and an Orbitrap Elite mass spectrometer (Thermo Electron Corp., San Jose, CA, USA). For phosphorylation site analysis samples were split into two portions—approximately 20% was subjected directly to LC/MS for standard peptide identification and the remaining 80% was subjected to enrichment of phosphorylated peptides on titanium dioxide, as described previously [54]. Briefly, peptides were diluted in 80% acetonitrile (ACN), 5% trifluoroacetic acid (TFA), 1 M phtalic acid and incubated with the titanium dioxide beads (GL Sciences, Tokyo, Japan). Non-phosphorylated peptides were washed away with 80% ACN, 0.1% TFA, then the phosphorylated ones were finally eluted by alkalization with ammonium hydroxide to pH 10.5.

Peptide mixture was applied to an RP-18 precolumn (nanoACQUITY Symmetry^®^ C18—Waters 186003514) using water containing 0.1% TFA as the mobile phase, and then transferred to a nano-HPLC RP-18 column (nanoACQUITY BEH C18—Waters 186003545) using an ACN gradient (0%–35% acetonitrile in 180 min) in 0.05% formic acid at a flow rate of 250 nL/min. The column outlet was coupled directly to the ion source of the spectrometer working in the regime of a data-dependent MS to MS/MS switch. A blank run ensuring the lack of cross contamination from previous samples preceded each analysis.

Raw data acquired were processed with a Mascot Distiller followed by a Mascot (Matrix Science, London, UK, on-site license) database search against the UniProt database, restricted to human sequences. The search parameters for the precursor and product ion mass tolerance were 30 ppm and 0.1 Da, respectively, enzyme specificity: semitrypsin, missed cleavage sites allowed: 1, fixed modification of cysteine by carbamidomethylation, and variable modifications of methionine oxidation and serine, threonine and tyrosine phosphorylation. Peptides with a Mascot Score exceeding the identity threshold calculated by Mascot were considered to be positively identified. Finally, all the peptides identified as phosphorylated were inspected manually.

### 4.12. FTO Dephosphorylation

Protein samples were mixed with experimental buffer DEP1 containing 800 U of Lambda Protein Phosphatase (#P0753S, New England BioLabs, Ipswich, MA, USA) to the concentration 3.5 µM in a final volume 50 µL and incubated for 30 min, at 30 °C with shaking (600 rpm).

### 4.13. Hydrogen–Deuterium Exchange (HDX)

^BES^FTO samples (40 µM) were incubated for 10 min at 4 °C in HDX1 incubation buffer alone or supplemented with 0.5 mM CaCl_2_ (HDX2). The samples were then diluted 10-fold with HDX3 reaction buffer (prepared in D_2_O). The hydrogen–deuterium exchange was terminated at: 0 s, 10 s, 1 min, 20 min, 1 h, and 24 h. The exchange was terminated by decreasing pH to app. 2–3 with HDX4 stop buffer and the samples were immediately frozen in liquid nitrogen. The extent of hydrogen–deuterium exchange was determined using a nanoACQUITY UPLC system (Waters Corporation, Milford, MA, USA). The samples, after initial protein digestion on a trypsin column, were analyzed by liquid chromatography coupled with mass spectrometry (LC-MS). Each experiment was repeated four times. The data were analyzed with the DynamX HDX Data Analysis Software 3.0 (Waters Corporation, Milford, MA, USA).

### 4.14. Gel Filtration Chromatography

Protein samples: 3.1 nmol of ^EC^FTO (7.8 µM) or 1.9 nmol of ^BES^FTO (4.7 µM) were incubated for 10 min at 4 °C in SEC1 buffer of final volume 400 µL and loaded onto a SEC650 column equilibrated and eluted with the SEC1 buffer. The eluate was monitored with NGC system ChromLab™ (BIO-RAD, Hercules, CA, USA) at 215, 260, and 280 nm. Protein standards (#1511901, BIO-RAD, Hercules, CA, USA) for calibration were analyzed by the same method. Due to UV-detection instead of iron we used manganium that does not require the presence of ascorbic acid.

### 4.15. Small-Angle X-ray Scattering (SAXS)

^BES^FTO (26 µM) was incubated for 5 min in SAXS1 buffer and then loaded into a low noise liquid sample cell (Xenocs, Grenoble, France). Samples were measured using an XEUSS 2.0 SAXS/WAXS system (Xenocs, Grenoble, France) with X-ray radiation (gallium K_α_ emission of 9.2 keV) produced by a MetalJet D2 microfocus generator (Excillum AB, Kista, Sweden). For each sample, 5-10 independent frames (exposition time 600 s per frame) were recorded using the PILATUS3 1M detector (DECTRIS Ltd., Baden-Daettwil, Switzerland). Scattering data were integrated using Foxtrot [55]. Data were analyzed using SCÅTTER software (http://www.bioisis.net/; accessed on 1 February 2020) and the PRIMUS program from the ATSAS package was used for curve fitting. Structural parameters (Rg Guinier and Rg p(r)) were estimated using the GNOM program from the ATSAS package [56]. Low resolution bead models were generated using the DAMMIN program [57]. The ensemble of FTO homodimer structures was obtained with the use of a model generated by the SymmDock server (http://bioinfo3d.cs.tau.ac.il/SymmDock/; accessed on 1 February 2020) [58].

## 5. Conclusions

Using two different FTO expression systems, bacterial and baculovirus, we identified three phosphoserines in the ^BES^FTO protein, all in the N-terminal domain. The two already known (S256 and S260) and one newly identified (S184). We were unable to determine unequivocally whether any of these phosphorylations affects the FTO catalytic activity; but the phosphomimetic substitution S256D clearly displayed decreased activity.

We confirmed the ability of FTO to dimerize via its C-terminal domain and the dimer dissociation constant turned out to be sufficiently low to permit FTO dimerization in the cell.

We identified differences in the biophysical properties of FTO from the two expression systems that, in spite of statistical significance were too small, suggesting that they may not be biologically relevant. Summarizing, both expression systems may be used interchangeably for experiments concerning the FTO protein alone.

## Figures and Tables

**Figure 1 ijms-22-04512-f001:**
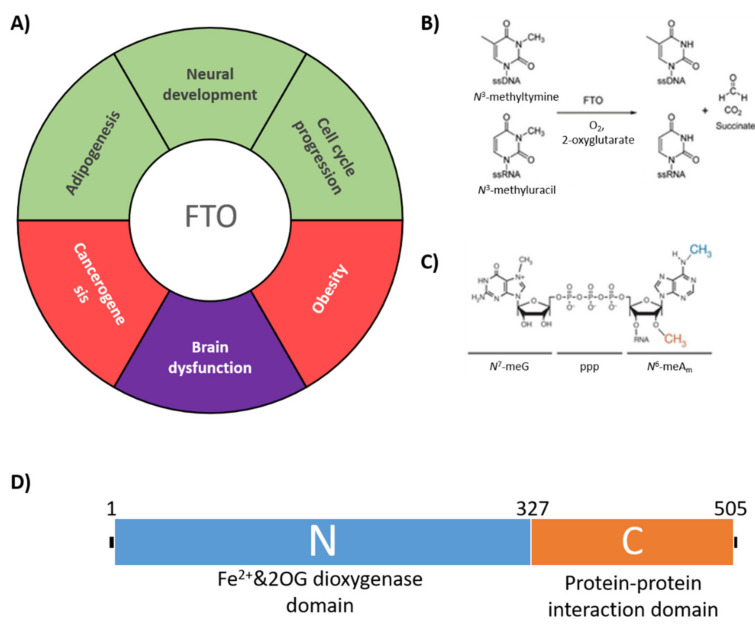
FTO architecture, specificity and functions. (**A**) Physiological functions affected by FTO. Green—physiological processes. Red—defects caused by FTO over-activity. Purple—defects caused by deficit of FTO activity; (**B**) demethylation reaction performed by FTO; (**C**) *N*^6^-meA_m_—the preferred FTO substrate (description in the text); (**D**) domain architecture of FTO protein.

**Figure 2 ijms-22-04512-f002:**
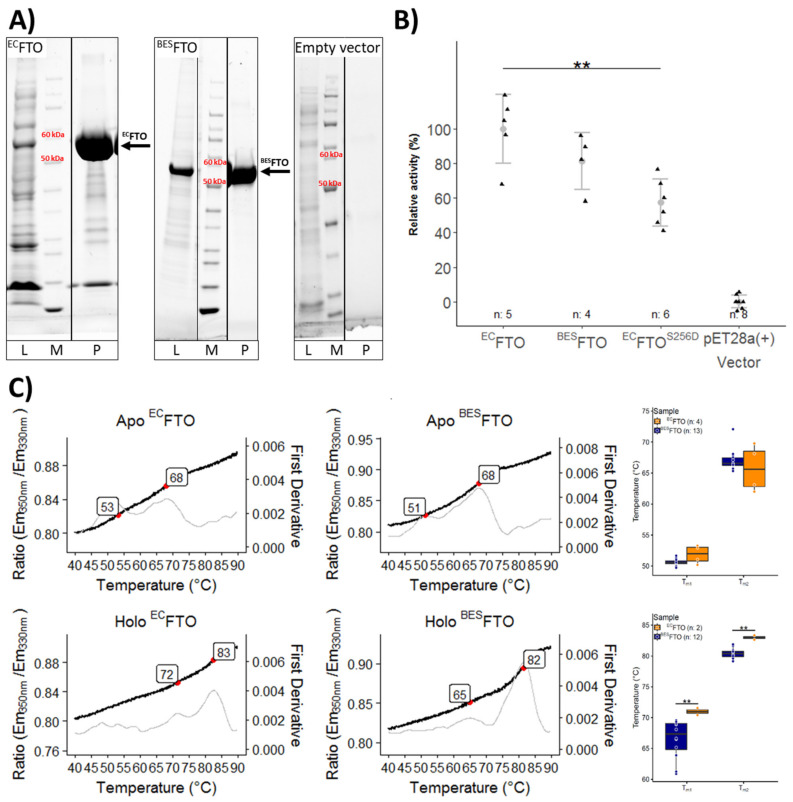
Purification and characterization of FTO proteins. (**A**) SDS-PAGE analysis of ^EC^FTO and ^BSE^FTO purification. *E. coli* BL21 carrying empty pET-28a(+) plasmid is a negative control. L—7.5 µL of cell lysate, M—molecular marker, P—purified ^EC^FTO (42 µg), ^BES^FTO (17 µg), or negative control. (**B**) Enzymatic activity of ^EC^FTO, ^BES^FTO and ^EC^FTO^S256D^ mutant based on chemiluminescence assay. *N*^6^-meA was used as a substrate. Relative activity is referenced to activity of 1 µg of ^EC^FTO (100%). Data are represented as points and means with whiskers showing standard deviation. n—number of samples. ** *p* < 0.05. (**C**) NanoDSF analysis of the thermal stability of purified ^EC^FTO and ^BES^FTO in the absence (apo) or presence of Mn^2+^ and 2-OG (holo). Left and middle, ratio of emissions (bold lines) and first derivative (thin lines) curves as a function of temperature. All four protein samples display two unfolding transitions (red dots): one (T_m1_) less prominent, at a lower temperature, and the second (T_m2_) clearer at a higher temperature. Right, gathered data represented as boxplots. The box represents the interquartile range. Whiskers show the highest and the lowest data points. The horizontal line within a box represents the median value. n—number of samples. ** *p* < 0.05.

**Figure 3 ijms-22-04512-f003:**
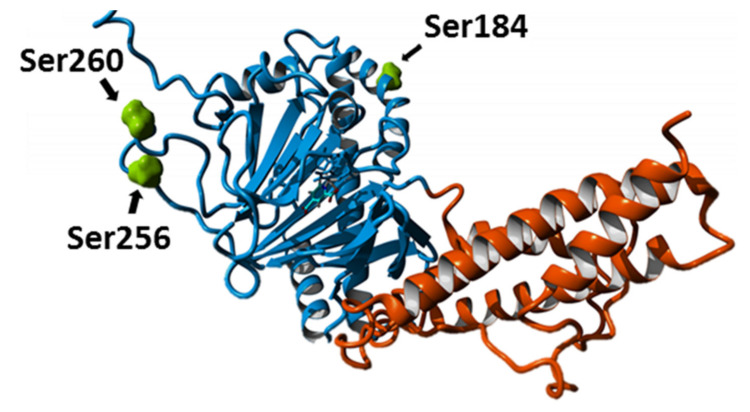
Location of phosphoserines identified in ^BES^FTO. The modifications (green) are located only in the N-terminal domain (blue); the C-terminal domain is in orange. The FTO protein structure was modelled basing on the PDB4IE5 record.

**Figure 4 ijms-22-04512-f004:**
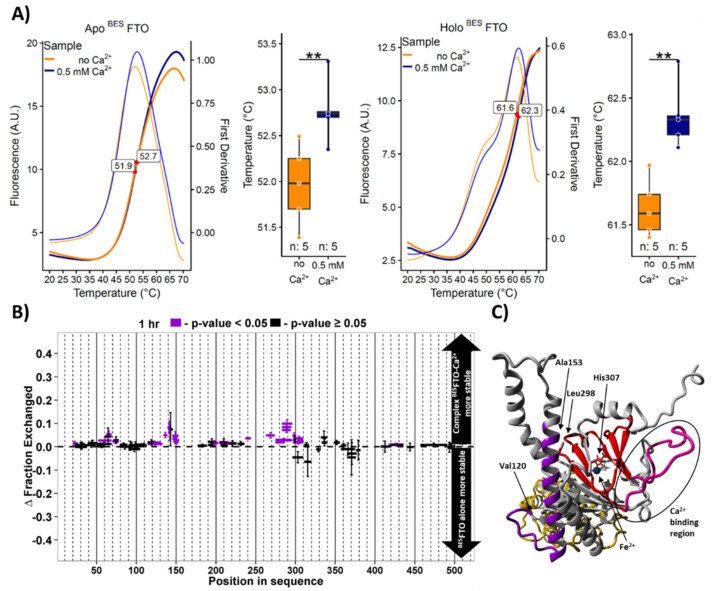
The effect of Ca^2+^ on FTO parameters. (**A**) DSF analysis of the Ca^2+^ effect on the thermal stability of ^BES^FTO in the absence (apo) or presence of Mn^2+^ and 2-OG (holo). Left, ratio of emissions (bold lines) and first derivative (thin lines) curves are shown as a function of temperature. All four protein samples display one distinct unfolding transition (red dots). Data gathered on the right are shown as boxplots. Box represents the interquartile range. Whiskers show the highest and lowest data points. The horizontal line within a box represents the median of the data. n—number of samples. A. U.—arbitrary units. ** *p* < 0.05. (**B**) HDX analysis of the Ca^2+^ effect on the solvent accessibility of ^BES^FTO after 1-h incubation with deuterium water. Differences between the exchange levels of individual peptides (each rectangle with whiskers represent one peptide) in samples with and without calcium are shown. Statistically significant differences (*p* < 0.05) in peptide accessibility are marked in purple. Two regions of ^BES^FTO are more stable in Ca^2+^ presence, comprising residues 120–155 and 240–310. Each experimental setup was repeated four times. (**C**) Location of Ca^2+^-stabilized regions in the FTO molecule colored according to HDX data. Regions with significant exchange are marked in red and purple; the Ca^2+^ binding part of the first region is marked in pink. The two protein regions stabilized by Ca^2+^ lie close to each other in the structure. The FTO protein structure was modelled based on the PDB4IE5 record.

**Figure 5 ijms-22-04512-f005:**
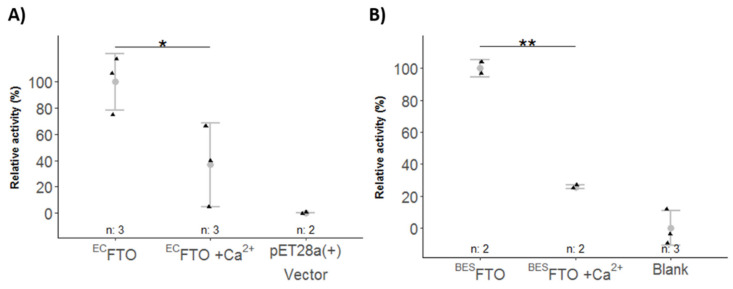
Effect of Ca^2+^ on enzymatic activity of FTO by chemiluminescence assay. Relative activity is referenced to the activity of 1 µg ^EC^FTO (**A**) or ^BES^FTO (**B**) (100%). The *N*^6^-meA was used as a substrate. Data are represented as points and means, with the whiskers showing sample standard deviation. n—number of samples. ** *p* < 0.05, * *p* < 0.1.

**Figure 6 ijms-22-04512-f006:**
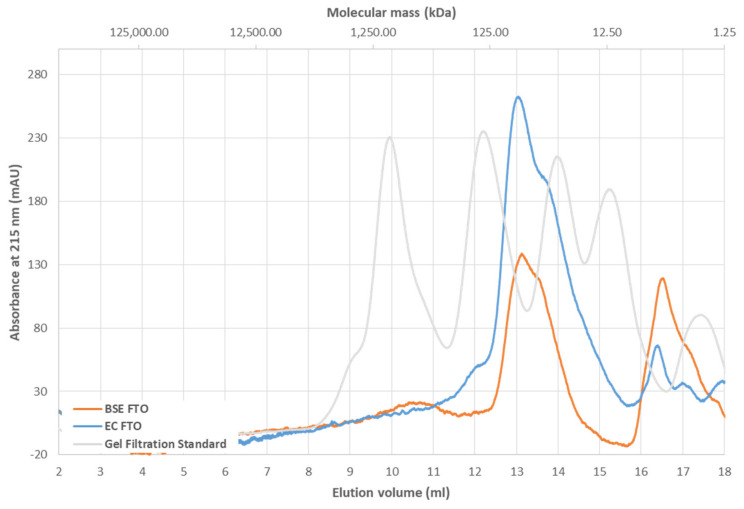
Gel filtration chromatography of ^EC^FTO (blue) and ^BES^FTO (orange) in the presence of 0.5 mM Mn^2+^ and 1 mM 2-OG. Chromatograms show the presence of both versions of FTO in fractions corresponding to two state of the protein: the one clearly monomeric (peak around 50 kDa) and the second one (peak around 80 kDa) where the dynamic dimer/monomer equilibrium occurs.

**Figure 7 ijms-22-04512-f007:**
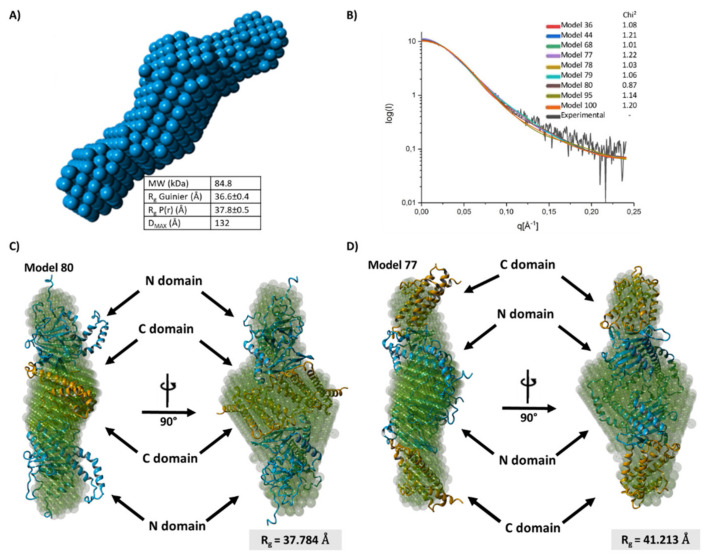
SAXS structural analysis of FTO homodimer. (**A**) Low resolution (~12–20 Å) model of the ^BES^FTO homodimer and its parameters obtained from SAXS data: MW—molecular weight of molecule, Rg Guinier—radius of gyration, Rg P(r)—radius pair distance distribution, Dmax—maximum dimension (**B**) Fitting of scattering data to ten best dimer structures generated in silico by SymmDock server. I—scattering intensity, Q—scattering vector. (**C**,**D**) The two best-fitting models of the FTO dimer, with monomers interacting via their C-terminal domains (**C**) or N-terminal domains (**D**), generated in silico with the SymmDock server (orange–blue ribbon structures) superimposed on the low resolution experimental structure (light blue).

**Figure 8 ijms-22-04512-f008:**
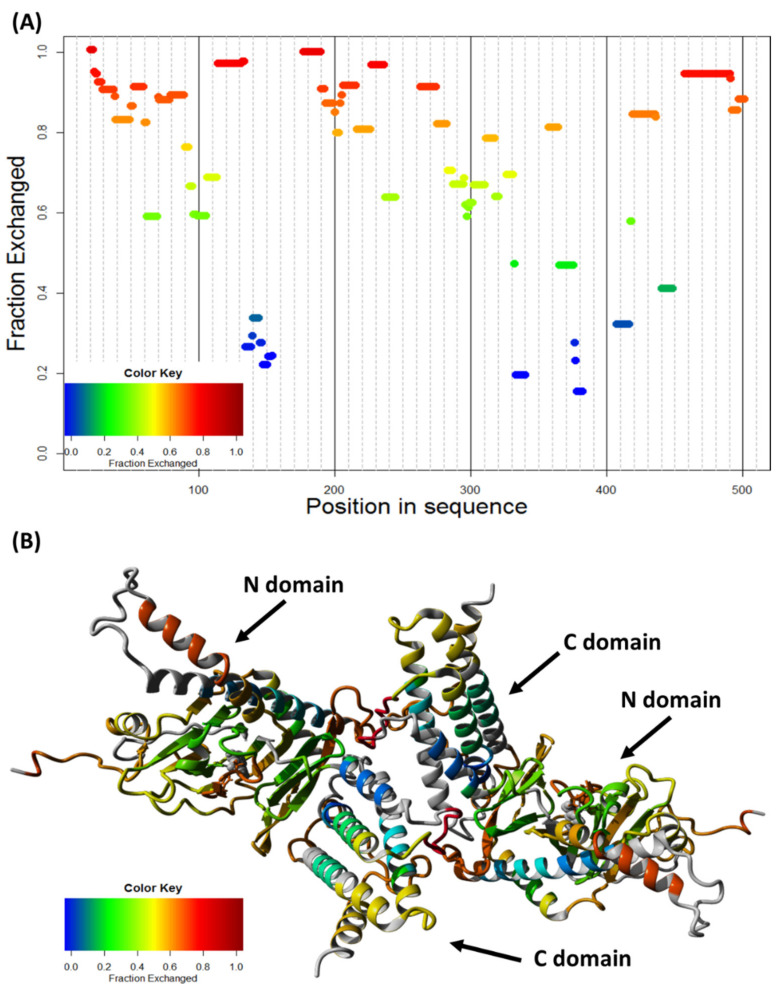
FTO dimer selection based on HDX data. (**A**) HDX profile of ^BES^FTO in the presence of Fe^2+^ and 2-OG, showing distribution of peptides (rectangles) colored according to the hydrogen exchange extent after 1 h. Two regions are substantially less solvent-accessible than the rest of the protein, residues 130–160 and 330–420. (**B**) Model of FTO homodimer connected through C-domains presented in Figure 7C, colored according to HDX exchange ratio.

**Figure 9 ijms-22-04512-f009:**
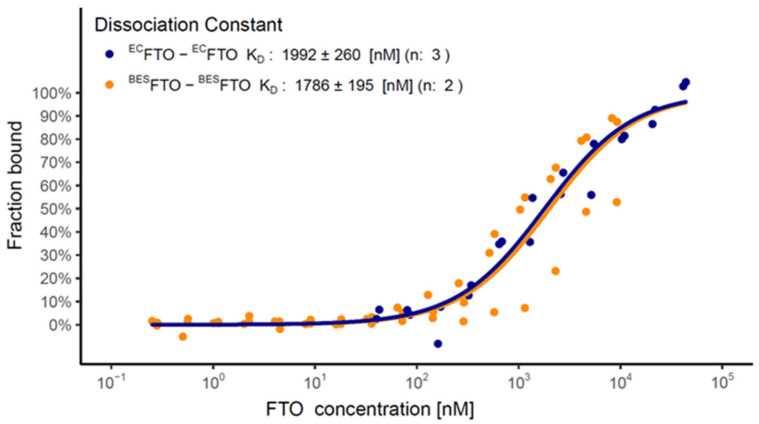
MST analysis of ^EC^FTO (blue) and ^BES^FTO (orange) dimerization in the presence of Fe^2+^ and 2-OG. The difference between the *K*_D_ values for ^EC^FTO and ^BES^FTO is not statistically significant. The plot represents the dimer level of the labeled protein at a given FTO concentration for each separate sample (data points) and for modeled equilibrium between monomer and dimer (straight lines). *K*_D_ are shown as estimated value ± standard deviation. n—number of repetitions.

**Table 1 ijms-22-04512-t001:** Compositions of buffers.

	Technique	Experiment	Buffer Composition
SDM1	Site Directed Mutagenesis	Preparation of FTO plasmids carrying point mutations S246G and S256D	50 mM Tris-HCl, pH 9.0, 5% (*v/v*) DMSO, 20 mM (NH_4_)_2_SO_4_, 2.5 mM MgCl_2_, 200 µM each dNTP, 0.5 µM each primer, 1 U of Marathon Polymerase (#1003-200, A&A Biotechnology), 0.8 ng of pET-28a(+) carrying hFTO cDNA
SOC	Bacteria culturing	Transformation	10 mM NaCl, 2.5 mM KCl, 2% (*w/v*) tryptone, 0.5% (*w/v*) yeast extract, 20 mM glucose, 10 mM MgCl_2_, 10 mM MgSO_4_
PRP1	Protein purification	Lysis bacteria	10 mM Na_2_HPO_4_, 1.8 mM KH_2_PO_4_, pH 7.4, 137 mM NaCl, 2.7 mM KCl, 10% (*v/v*) glycerol, 0.1% (*v/v*) TWEEN^®^ 20, 0.1% (*w/v*) lysozyme, 5 mM imidazole, 5 mM β-mercaptoethanol,
PRP2	Protein purification	Elution from Ni-charged Resin	10 mM Na_2_HPO_4_, 1.8 mM KH_2_PO_4_, pH 7.4, 137 mM NaCl, 2.7 mM KCl, 10% (*v/v*) glycerol, 0.1% (*v/v*) TWEEN^®^ 20, 150 mM imidazole, 5 mM β-mercaptoethanol,
PRP3	Protein purification	Dialysis	50 mM Tris-HCl, pH 7.5, 150 mM NaCl, 50% (*v/v*) glycerol, 5mM β-mercaptoethanol
PRP4	Protein purification	Lysis of BES insect cells	50 mM HEPES, pH 7.5, 150 mM NaCl, 10% (*v/v*) glycerol, 5 mM β-mercaptoethanol
TSA1	Differential Scanning Fluorimetry	Evaluation of apo ^BES^FTO thermal stability	50 mM HEPES, pH 7.5, 150 mM NaCl, 1% (*v/v*) SYPRO^®^ ORANGE stock (#S6650, Life Technologies), 2 mM L-ascorbic acid
TSA2	Differential Scanning Fluorimetry	Evaluation of effect of calcium on apo ^BES^FTO thermal stability	50 mM HEPES, pH 7.5, 150 mM NaCl, 1% (*v/v*) SYPRO^®^ ORANGE stock, 2 mM L-ascorbic acid, 0.5 mM CaCl_2_
TSA3	Differential Scanning Fluorimetry	Evaluation of holo ^BES^FTO thermal stability	50 mM HEPES, pH 7.5, 150 mM NaCl, 1% (*v/v*) SYPRO^®^ ORANGE stock, 2 mM L-ascorbic acid, 1 mM 2-OG, 0.5 mM (NH_4_)_2_Fe(SO_4_)_2_
TSA4	Differential Scanning Fluorimetry	Evaluation of effect of calcium on holo ^BES^FTO thermal stability	50 mM HEPES pH 7.5, 150 mM NaCl, 1% (*v/v*) SYPRO^®^ ORANGE stock and 2 mM L-ascorbic acid, 1 mM 2-OG, 0.5 mM (NH_4_)_2_Fe(SO_4_)_2_, 0.5 mM CaCl_2_
DSF1	Nano Differential Scanning Fluorimetry	Evaluation of apo ^EC^FTO and apo ^BES^FTO thermal stability	50 mM MES, pH 6.5, 150 mM NaCl, 10% (*v/v*) glycerol
DSF2	Nano Differential Scanning Fluorimetry	Evaluation of effect of Mn^2+^ on ^BES^FTO thermal stability	50 mM MES, pH 6.5, 150 mM NaCl, 10% (*v/v*) glycerol, 0.5 mM MnCl_2_
DSF3	Nano Differential Scanning Fluorimetry	Evaluation effect of 2-OG on ^BES^FTO thermal stability	50 mM MES, pH 6.5, 150 mM NaCl, 10% (*v/v*) glycerol, 1mM 2-OG
DSF4	Nano Differential Scanning Fluorimetry	Evaluation of holo ^EC^FTO and holo ^BES^FTO thermal stability	50 mM MES, pH 6.5, 150 mM NaCl, 10% (*v/v*) glycerol, 1 mM 2-OG, 0.5 mM MnCl_2_,
DSF5	Nano Differential Scanning Fluorimetry	Evaluation of effect of dephosphorylation on ^EC^FTO and ^BES^FTO thermal stability	37 mM HEPES, 1 mM MES, pH 7.5, 124 mM NaCl, 37% (*v/v*) glycerol, 0.01% Brij 35, 3.7 mM β-mercaptoethanol, 2 mM DTT, 1 mM 2-OG, 1 mM Mn Cl_2_
DSF6	Nano Differential Scanning Fluorimetry	Evaluation effect of dephosphorylation on ^EC^FTO and ^BES^FTO thermal stability	37 mM HEPES, 1mM MES, pH 7.5, 124 mM NaCl, 37% (*v/v*) glycerol, 0.01% Brij 35, 3.7 mM β-mercaptoethanol, 2 mM DTT, 1 mM 2-OG, 1 mM Mn Cl_2_, 800 U of Lambda Protein Phosphatase (#P0753S, New England BioLabs)
DEP1	Dephosphorylation	Dephosphorylation of ^EC^FTO and ^BES^FTO	37 mM HEPES, 1mM MES, pH 7.5, 150 mM NaCl, 37% (*v/v*) glycerol, 3.7 mM β-mercaptoethanol, 1 mM 2-OG, 1 mM MnCl_2_, 1x NEBuffer for Protein MetalloPhosphatases (#B0761S, New England BioLabs)
HDX1	Hydrogen–Deuterium Exchange	Sample preincubation	50 mM MES, pH 6.1, 150 mM NaCl, 0.004% (*v/v*) TWEEN^®^ 20, 2 mM L-ascorbic acid, 1 mM 2-OG, 0.5 mM (NH_4_)_2_Fe(SO_4_)_2_
HDX2	Hydrogen–Deuterium Exchange	Sample preincubation with Ca^2+^	50 mM MES pH 6.1, 150 mM NaCl, 0.004% (*v/v*) TWEEN^®^ 20, 2 mM L-ascorbic acid, 1 mM 2-OG, 0.5 mM (NH_4_)_2_Fe(SO_4_)_2_, 0.5 mM CaCl_2_
HDX3	Hydrogen–Deuterium Exchange	HDX reaction	30 mM Tris-DCl, pD 7.5 (pH_READ_+0.4), 150 mM NaCl in D_2_O (#DLM-4DR-99.8-PK, Cambridge Isotope Laboratories, Inc.)
HDX4	Hydrogen–Deuterium Exchange	HDX stopping	2 M glycine, pH 2.4, 107 mM NaCl
SEC1	Gel Filtration Chromatography	Estimation of molecular size of protein	50 mM MES, pH 6.5, 150 mM NaCl, 10% (*v/v*) glycerol_,_ 1 mM 2-OG, 0.5 mM MnCl_2_
MST1	Microscale thermophoresis	Evaluation of dimerization and interaction with Fe^2+^ and 2-OG of ^BES^FTO	50 mM MES, pH 6.1, 150 mM NaCl, 0.05% (*v/v*) TWEEN^®^ 20, 2 mM L-ascorbic acid
MST2	Microscale thermophoresis	Evaluation of Fe^2+^ effect on dimerization and interaction with 2-OG of ^BES^FTO	50 mM MES, pH 6.1, 150 mM NaCl, 0.05% (*v/v*) TWEEN^®^ 20, 2 mM L-ascorbic acid, 0.5 mM (NH_4_)_2_Fe(SO_4_)_2_
MST3	Microscale thermophoresis	Evaluation of 2-OG effect on dimerization and interaction with Fe^2+^ of ^BES^FTO	50 mM MES pH 6.1, 150 mM NaCl, 0.05% (*v/v*) TWEEN^®^ 20, 2 mM L-ascorbic acid, 1 mM 2-OG
MST4	Microscale thermophoresis	Evaluation of dimerization of ^EC^FTO and ^BES^FTO	50 mM MES pH 6.1, 150 mM NaCl, 0.05% (*v/v*) TWEEN^®^ 20, 2 mM L-ascorbic acid, 1 mM 2-OG, 0.5 mM (NH_4_)_2_Fe(SO_4_)_2_
SAX1	Small-angle X-ray Scattering	Evaluation of holo ^BES^FTO structure in water solution	33 mM HEPES, 13 mM MES, pH 7.0, 150 mM NaCl, 7.3% (*v/v*) glycerol, 3.3 mM β-mercaptoethanol, 2 mM ascorbic acid, 1 mM 2-OG, 0.5 mM (NH_4_)_2_Fe(SO_4_)_2_, 0.5 mM CaCl_2_

## Data Availability

The data presented in this study are available on request from the corresponding authors.

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
