# Peer review of "Effect of Posttranslational Modifications on the Structure and Activity of FTO Demethylase"

_ijms, 2021, doi:10.3390/ijms22094512_

Round 1
Reviewer 1 Report
FTP (FaT mass and Obesity-associated) protein is a demethylase where its N-terminal domain is the catalytic center and C-terminal domain maintains the catalytic domain in a proper structure. FTO is involved in a wide range of key physiological processes. Dysfunction of FTO leads to cancers and improper brain development. Catalytic reaction of FTO occurred in the presence of O2 and Fe2+ as cofactors and 2-OG as a co-substrate.
Few reports about the relationship between FTO structure and function. There were some reports about the phosphorylated residues influencing the FTO life time, as well as cellular localization and its substrate specificity. The author analyzed the relationship between structure and activity of FTO and the effects of diverse modulating factors. Particularly, they determined the influences of the FTO phosphorylation status, small molecules and ions on its properties from two expression system--- E.coli (EC) expression system versus baculovirus (BES) expression system, in which posttranslational modifications are supposed to be preserved.
Overall, this is well-written and comprehensive, clearly demonstrated the relationship between structural and functional aspects.
Some suggestions below:
- Role of S184 phosphorylation was presumed to enhance the interactions with the proximal basic residues (K162 and R178) and thus, stabilizes a particular conformation. It would be worth show direct data that S184 phosphorylation is somehow affecting the stability. The role of this phosphorylated residue is not clear here.
- Ca2+ is presumed to coordinate with Fe2+ to modulate demethylase activity; the three phosphorylated sites were located on the putative Ca2+ binding site whereas phosphorylation of S256 and S260 contribute to Ca2+ binding. How does Ca2+ coordinate with Fe2+ to decrease catalytic activity?
- They did well explanations in the Discussion about why the proteins from two expression system showed no significantly difference in the enzymatic activity since the phosphorylation play roles in it.
- Discussion also mentioned that “a sequence events with phosphorylation initially decreasing FTO activity followed by its degradation by the cellular machinery”. It would be good to show that S184D, S260D single mutation or double or triple mutations (or else) are affecting the FTO activity and stability for the future study.
- Discussion, line 371, there is a typo “lover”, which should be “lower”.
Author Response
Response to Reviewer 1 Comments
Point 1: Role of S184 phosphorylation was presumed to enhance the interactions with the proximal basic residues (K162 and R178) and thus, stabilizes a particular conformation. It would be worth show direct data that S184 phosphorylation is somehow affecting the stability. The role of this phosphorylated residue is not clear here.
Response 1: As it is stated in the text, we performed molecular modelling to analyse how phosphorylation of S184 affects its interaction with the potentially proximal basic residues (K162 and R178), thereby stabilizing a particular conformation of the protein. At the moment we cannot perform the required experiment, but following the reviewer suggestions we made the alternative in silico analysis of the protein stability using dedicated server (RaptorX). The analysis clearly showed that the partially disordered loop (N164-G187), which is invisible in the FTO PDB structures is significantly stabilized by S184D and S184E replacements. Interestingly, a distant region of the protein (D347-N372) is also minutely affected, however this phenomenon requires further experimental investigations. We are still working on other phosphorylation-mimicking replacements in ECFTO, but at the moment we can show solely analyses of the most abundant one.
The appropriate explanation has been introduce in the text (lines 178-182), and the results of in silico analyses are shown in Supplementary Figure 4.
Point 2: Ca2+ is presumed to coordinate with Fe2+ to modulate demethylase activity; the three phosphorylated sites were located on the putative Ca2+ binding site whereas phosphorylation of S256 and S260 contribute to Ca2+ binding. How does Ca2+ coordinate with Fe2+ to decrease catalytic activity?
Response 2: Sorry for the misunderstanding – Ca2+ and Fe2+ binding sites are distant. Now, we explained it better in the text (lines 227-229). Ca2+is not coordinating Fe2+ directly, but interacts at quite different protein site. The residues affected by Ca2+ interaction (HDX results) are located in such way that the part of the catalytic site and the region surrounding Ca2+-binding loop seem to stick to each other. This is the putative mechanism coupling the FTO catalytic activity with the presence of Ca2+. Please also note that only two out of three phosphorylations are putatively involved in Ca2+ interaction.
Point 3: They did well explanations in the Discussion about why the proteins from two expression system showed no significantly difference in the enzymatic activity since the phosphorylation play roles in it.
Response 3: Thank you!
Point 4: Discussion also mentioned that “a sequence events with phosphorylation initially decreasing FTO activity followed by its degradation by the cellular machinery”. It would be good to show that S184D, S260D single mutation or double or triple mutations (or else) are affecting the FTO activity and stability for the future study.
Response 4: No doubt, the Reviewer is right that double or even triple mutants should be investigated. We are working on such mutants, however, at the moment we were succeeded solely with S256D replacement. We attempted to obtain S184D and S260D, but some undesired additional mutations have been identified. However, we thank for this remark and we will continue our work on it.
Point 5: Discussion, line 380, there is a typo “lover”, which should be “lower”.
Response 5: Has been corrected.

Reviewer 2 Report
This manuscript analyzed the activity of the protein obtained by expressing FTO demethylase in the E. coli and baculovirus systems. Functional analysis were performed using enzyme activity, nanoDSF, MST and SAXS. This study is an extension of the work of previous authors and is expected to contribute to related fields. However, in order for the manuscript to be recommended for publication, it is necessary to clarify several contents as below.
- Due to the function of FTO demethylase, Fe2+ is considered to be a functionally necessary metal. In the nanoDSF experiment, Mn2+ was used instead of Fe2+. How abou the nanoDSFresult when using Fe2+? The author mentioned "Mn2+-non-catalytic cation mimicking Fe2+". Can the author be sure that Mn2+ binds only to the same position as Fe2+ in FTO?
- The author has made a lot of comments on the activity of S256 and S260. Meanwhile, these sites are far from active sites. How do you consider it to contribute to the activity? Meanwhile, ECFTO has higher enzymatic activity compared to BESFTO. Can you conclude that posttranslational modification inhibited activity?
- line398-400: "On the other hand, Han and coworkers obtained a crystallographic structure with FTO in a trimeric form [18], proposed that the trimer could only be found in FTO crystals." This seems to have been misunderstood by the author. Previous research did not mention that this molecule is a timer.
"In crystal, a trimer related by crystallographic three-fold symmetry is mainly mediated by Trp 278 (Supplementary Fig. 2). However, the mutation Trp278Asp had no effect on its position under gel-filtration assay and its enzymatic activity (data not shown), indicating that such a trimer resulted from crystal packing." doi: 10.1038/nature08921
- The authors commented on the effects of Ca2+ on the stability and activity of FTO. There is no comparison to be considered as an intrinsic property for Ca2+. What about other metal ions?
- This paper mentions phosphoserine residues (S256, S260 and S184), of which S184 (3%) is novel compared to previous studies, but not much has been covered.
Minor
- line 27: "Mn 2+" should be "Mn2+"
- line 174: "PDB" should be "Protein Data Bank". Author should added the PDB code.
- line 491: "Na-EDTA" is unclear.
- line 621: "supplemented with 0.5 mM CaCl2 (HDX2)," a period at the end of the sentence.
- I suggest that Table 1 moves to supplementary data.
- There are many typos in Table 2. And there is inconsistency in marking the buffer. It should be revised and unified.
- I sugest athat amino acid sequences moves to the supplementary data.
- The PDB code for Figure 3 and Figure 4D should be added to the figure caption.
- The resolution of Figure 8 should be increased and removes the gray lines in the background.
Author Response
Response to Reviewer 2 Comments
Point 1: Due to the function of FTO demethylase, Fe2+ is considered to be a functionally necessary metal. In the nanoDSF experiment, Mn2+ was used instead of Fe2+. How abou the nanoDSFresult when using Fe2+? The author mentioned "Mn2+-non-catalytic cation mimicking Fe2+". Can the author be sure that Mn2+ binds only to the same position as Fe2+ in FTO?
Response 1: Fe2+ is crucial for the FTO function, but not for the structure. Only in 3 structures of FTO accessible in PDB the Fe2+ is present at the catalytic site (3lfm,5dab,5f8p), while in the next 15 Fe2+ is replaced by the isostructural Mn2+ (4qkn, 4zs2, 4zs3, 5zmd), Ni2+ (4cxw, 4cxx, 4cxy, 4cxy, 4idz, 6eoz) or Zn2+ (4ie0, 4ie4, 4ie5, 4ie6, 4ie7, 4qho, 6aej ,6ak4), all of which bind at the same site. Since reliable experiment with the use of Fe2+ is challenging due to spontaneous Fe2+ to Fe3+oxidation at pH 6-8, we decide to use Mn2+ in all structure-related experiments. This was especially important for nanoDSF studies. In all pdb structures of FTO the only Mn-binding site in that at which Fe2+ binds. Mn2+ can putatively bind to a His-tag, however this interaction should not interfere with the protein stability. Expanded explanation has been attached to the first paragraph of the 5.8 section (lines 538-541).
Point 2: The author has made a lot of comments on the activity of S256 and S260. Meanwhile, these sites are far from active sites. How do you consider it to contribute to the activity? Meanwhile, ECFTO has higher enzymatic activity compared to BESFTO. Can you conclude that posttranslational modification inhibited activity?
Response 2: The results from HDX experiments indicated that factors affecting loop containing the two mentioned residues, are proximal to residues that are important for FTO catalytic activity, what is pointed out more clearly in the text in the lines 227-229, 238 and 240 and in the modified Figure 4C. We noticed the decrease of the catalytic activity of BESFTO relative to ECFTO, but the difference was not statistically significant, which can be explained by relative low level of the observed posttranslational modifications. This fact was a premise to measure the activity of S256D variant of ECFTO that mimicked the phosphorylation of this residue. Lower catalytic activity of the ECFTOS256D in comparison to ECFTO clearly indicates that this particular posttranslational modification decreases the enzymatic activity. The effect of the other two phosphorylations is still an open question that we plan to investigate.
Point 3: Line 398-400: "On the other hand, Han and coworkers obtained a crystallographic structure with FTO in a trimeric form [18], proposed that the trimer could only be found in FTO crystals." This seems to have been misunderstood by the author. Previous research did not mention that this molecule is a timer.
"In crystal, a trimer related by crystallographic three-fold symmetry is mainly mediated by Trp 278 (Supplementary Fig. 2). However, the mutation Trp278Asp had no effect on its position under gel-filtration assay and its enzymatic activity (data not shown), indicating that such a trimer resulted from crystal packing." doi: 10.1038/nature08921
Response 3: We are grateful to the Reviewer for pointing this inconsistency. We corrected lines 408-410.
Point 4: The authors commented on the effects of Ca2+ on the stability and activity of FTO. There is no comparison to be considered as an intrinsic property for Ca2+. What about other metal ions?
Response 4: It was hard to investigate other biologically relevant ions due to already known interaction with the other parts of examined protein. Fe2+ Mn2+, Ni2+ and Zn2+ binds to the catalytic core of the active protein. Mn2+, Cu2+, Ni2+ Zn2+ and Co2+ interact with the HisTag (Terpe, 2003). Thus, we focused on the particular interaction with the Ca2+ only, due to the high probability of the mentioned disordered loop binds calcium specifically (Rigden et al., 2004 and Santamaria-Hernando et al., 2012).
Point 5: This paper mentions phosphoserine residues (S256, S260 and S184), of which S184 (3%) is novel compared to previous studies, but not much has been covered.
Response 5: As we stated in the text, we performed molecular modelling predicting that phosphorylation of S184 enhance interaction of this amino acid with the proximal basic residues (K162 and R178), thereby stabilizing a particular conformation of protein. We also investigated in silico which region of FTO N-terminal domain showing highest probability to be disordered and how single point mutations mimicking phosphorylation state, affects protein disordering. The appropriate explanation has been introduce in the text (lines 178-182), and the results of in silico analyses are shown in Supplementary Figure 4.
Point 6: line 27: "Mn 2+" should be "Mn2+".
Response 6: Improved.
Point 7: line 174: "PDB" should be "Protein Data Bank". Author should added the PDB code.
Response 7: "PDB" have been replaced by "Protein Data Bank" (lines 174 and 509); missing PDB codes have been included (lines 188 and 255).
Point 8: line 502: "Na-EDTA" is unclear.
Response 8: It was exchanged into “EDTA”.
Point 9: line 636: "supplemented with 0.5 mM CaCl2 (HDX2)," a period at the end of the sentence.
Response 9: Improved.
Point 10: I suggest that Table 1 moves to supplementary data.
Response 10: We done it.
Point 11: There are many typos in Table 2. And there is inconsistency in marking the buffer. It should be revised and unified.
Response 11: We improved consistency of the Table 2.
Point 12: I sugest athat amino acid sequences moves to the supplementary data.
Response 12: We done it.
Point 13: The PDB code for Figure 3 and Figure 4D should be added to the figure caption.
Response 13: Improved.
Point 14: The resolution of Figure 8 should be increased and removes the gray lines in the background.
Response 14: Improved.

Round 2
Reviewer 2 Report
The author has addressed some concerns and the manuscript has been improved. However, there are several other issues with manuscripts for publication.
Point 1: nanoDSF with Fe2+.
Author Response 1: Fe2+ is crucial for the FTO function, but not for the structure. Only in 3 structures of FTO accessible in PDB the Fe2+ is present at the catalytic site (3lfm,5dab,5f8p), while in the next 15 Fe2+ is replaced by the isostructural Mn2+ (4qkn, 4zs2, 4zs3, 5zmd), Ni2+ (4cxw, 4cxx, 4cxy, 4cxy, 4idz, 6eoz) or Zn2+ (4ie0, 4ie4, 4ie5, 4ie6, 4ie7, 4qho, 6aej ,6ak4), all of which bind at the same site.
█ In the paper on the Mn2+, Ni2+ and Zn2+ bound FTO structure, has it been confirmed that the metal included in the model is correct? This is because the metal sites of some crystal structures are not verified and make the wrong model.
Author Response 1:Since reliable experiment with the use of Fe2+ is challenging due to spontaneous Fe2+ to Fe3+oxidation at pH 6-8, we decide to use Mn2+ in all structure-related experiments. This was especially important for nanoDSF studies.
█ According to the author's claim, the nanoDSF problem may be solved, but in another experiment (e.g. MST experiment, pH 6-8), it is said that the experiment was performed with Fe3+ or Fe2+/Fe3+ mixture instead of Fe2+. In other experiments, is there any effect even if Fe3+ or Fe2+/Fe3+ mixture is used? And isn't there a problem with describing Fe2+ in manuscript?
█ On the other hand, does the author judge that both Fe2+ and Fe3+ can bind to the catalytic site? If only Fe2+ is selectively bound, then the Fe3+ conversion will not be a problem in the experiment because it is in the solution without binding to the protein.
Point 2: Ca2+ affects FTO catalytic activity
Line61-67: “Although the stabilizing effect of Ca2+ was rather small, it affected regions of the N-terminal domain of BESFTO involved in enzymatic activity. Indeed, Ca2+ was found to decrease the activity of both FTO preparations (Figure 5A,B). Despite its marginal effect on FTO stability, Ca2+ decreased the demethylase activity significantly, in the case of BESFTO by more than 80%. Moreover, the stronger effect observed for BESFTO agreed with our in silico and biophysical predictions that phosphoserines 256 and 260 contribute to Ca2+ binding.
█ As mentioned above, the activity of ECFTO and BESFTO was decreased by Ca2+. Meanwhile, the author said, “our in silico and biophysical predictions that phosphoserines 256 and 260 contribute to Ca2+ binding. Insisted. There are no phosphosrines in ECFTO, how do I understand it?
I suggest experimenting with other metals as well to address these overall uncertain results.
Author Response
Dear reviewer,
We would like to thank you for your in-depth examination of our text and apologize for any inaccuracies. Please find a more comprehensive explanation of your concerns below.
Point 1: nanoDSF with Fe2+.
Author Response 1: Fe2+ is crucial for the FTO function, but not for the structure. Only in 3 structures of FTO accessible in PDB the Fe2+ is present at the catalytic site (3lfm,5dab,5f8p), while in the next 15 Fe2+ is replaced by the isostructural Mn2+ (4qkn, 4zs2, 4zs3, 5zmd), Ni2+ (4cxw, 4cxx, 4cxy, 4cxy, 4idz, 6eoz) or Zn2+ (4ie0, 4ie4, 4ie5, 4ie6, 4ie7, 4qho, 6aej ,6ak4), all of which bind at the same site.
█ In the paper on the Mn2+, Ni2+ and Zn2+ bound FTO structure, has it been confirmed that the metal included in the model is correct? This is because the metal sites of some crystal structures are not verified and make the wrong model.
Response 1: Manganium is commonly used as the non-catalytic equivalent of Fe2+ in structural studies on dioxygenases belonging to AlkB family. The UNWANTED enzymatic activity may lead to product appearance preventing crystals formation, so Fe(II) is to be replaced by other isostructural bivalent ions, most commonly Mn2+. Structures of proteins other than FTO belonging to AlkB family dioxygenases containing manganium ion could be found for example in:
- Crystal structures of DNA/RNA repair enzymes AlkB and ABH2 bound to dsDNA Yang, C.-G., Yi, C., Duguid, E.M., Sullivan, C.T., Jian, X., Rice, P.A., He, C. (2008) Nature 452: 961-965
· Iron-catalysed oxidation intermediates captured in a DNA repair dioxygenase.Yi, C., Jia, G., Hou, G., Dai, Q., Zhang, W., Zheng, G., Jian, X., Yang, C.G., Cui, Q., He, C. (2010) Nature 468: 330-333
· Duplex interrogation by a direct DNA repair protein in search of base damage Yi, C., Chen, B., Qi, B., Zhang, W., Jia, G., Zhang, L., Li, C.J., Dinner, A.R., Yang, C.G., He, C. (2012) Nat Struct Mol Biol 19: 671-676
· Identification and analysis of adenine N6-methylation sites in the rice genome. Zhou, C., Wang, C., Liu, H., Zhou, Q., Liu, Q., Guo, Y., Peng, T., Song, J., Zhang, J., Chen, L., Zhao, Y., Zeng, Z., Zhou, D.X. (2018) Nat Plants 4: 554-563
· Structural basis of nucleic acid recognition and 6mA demethylation by human ALKBH1. Tian, L.F., Liu, Y.P., Chen, L., Tang, Q., Wu, W., Sun, W., Chen, Z., Yan, X.X. (2020) Cell Res 30: 272-275
· Switching Demethylation Activities between AlkB Family RNA/DNA Demethylases through Exchange of Active-Site Residues. Zhu, C., Yi, C. (2014) Angew Chem Int Ed Engl 53: 3659-3662
· Enzymological and structural studies of the mechanism of promiscuous substrate recognition by the oxidative DNA repair enzyme AlkB. Yu, B., Hunt, J.F. (2009) Proc Natl Acad Sci U S A 106: 14315-14320
· Crystal structure and RNA binding properties of the RNA recognition motif (RRM) and AlkB domains in human AlkB homolog 8 (ABH8), an enzyme catalyzing tRNA hypermodification. Pastore, C., Topalidou, I., Forouhar, F., Yan, A.C., Levy, M., Hunt, J.F. (2012) J Biol Chem 287: 2130-2143
· The atomic resolution structure of human AlkB homolog 7 (ALKBH7), a key protein for programmed necrosis and fat metabolism Wang, G., He, Q., Feng, C., Liu, Y., Deng, Z., Qi, X., Wu, W., Mei, P., Chen, Z. (2014) J Biol Chem 289: 27924-27936
· Crystal structures of the human RNA demethylase Alkbh5 reveal basis for substrate recognition Feng, C., Liu, Y., Wang, G., Deng, Z., Zhang, Q., Wu, W., Tong, Y., Cheng, C., Chen, Z. (2014) J Biol Chem 289: 11571-11583
· 5-Carboxy-8-hydroxyquinoline is a Broad Spectrum 2-Oxoglutarate Oxygenase Inhibitor which Causes Iron Translocation. Hopkinson, R.J., Tumber, A., Yapp, C., Chowdhury, R., Aik, W., Che, K.H., Li, X.S., Kristensen, J.B.L., King, O.N.F., Chan, M.C., Yeoh, K.K., Choi, H., Walport, L.J., Thinnes, C.C., Bush, J.T., Lejeune, C., Rydzik, A.M., Rose, N.R., Bagg, E.A., McDonough, M.A., Krojer, T., Yue, W.W., Ng, S.S., Olsen, L., Brennan, P.E., Oppermann, U., Muller-Knapp, S., Klose, R.J., Ratcliffe, P.J., Schofield, C.J., Kawamura, A. (2013) Chem Sci 4: 3110-3117
· Crystal structure of the RNA demethylase ALKBH5 from zebrafish. Chen, W., Zhang, L., Zheng, G., Fu, Y., Ji, Q., Liu, F., Chen, H., He, C. (2014) FEBS Lett 588: 892-898
· Rhein Inhibits AlkB Repair Enzymes and Sensitizes Cells to Methylated DNA Damage. Li, Q., Huang, Y., Liu, X., Gan, J., Chen, H., Yang, C.G.(2016) J Biol Chem 291: 11083-11093
Moreover, the structures containing Mn2+ (4qkn), Ni2+ (4cxx, 4idz) and Zn2+ (4ie5, 4ie6) were verified by the CheckMyMetal server (Zheng, H CheckMyMetal: a macromolecular metal-binding validation tool (2017) Acta crystallographica. Section D, Structural biology, 73: 223-233; DOI: 10.1107/S2059798317001061), that confirmed correctness of the models. In all cases, mentioned metals occupied Fe2+ binding site (Figure 1).
Additionally, we evaluate software, through verification model 4ie5 where Zn2+ were substituted by selected metal ions: Fe2+, Ca2+, Mg2+, Na+, K+ (Figure 2). Only FTO model containing Fe2+ in catalytic centre possesses acceptable parameters assigned by CheckMyMetal.
Point 2:
Author Response 1:Since reliable experiment with the use of Fe2+ is challenging due to spontaneous Fe2+ to Fe3+oxidation at pH 6-8, we decide to use Mn2+ in all structure-related experiments. This was especially important for nanoDSF studies.
█ According to the author's claim, the nanoDSF problem may be solved, but in another experiment (e.g. MST experiment, pH 6-8), it is said that the experiment was performed with Fe3+ or Fe2+/Fe3+ mixture instead of Fe2+. In other experiments, is there any effect even if Fe3+ or Fe2+/Fe3+ mixture is used? And isn't there a problem with describing Fe2+ in manuscript?
Response 2: We are sorry that our previous explanation was not precise enough. We used Mn2+ instead of Fe2+ solely in experiments based on direct UV detection, in which the presence of ascorbic acid would interfere with UV detection (i.e. label-free nanoDSF and size-exclusion chromatography). We have added appropriate explanation in lines 568-576.
Point 3:
█ On the other hand, does the author judge that both Fe2+ and Fe3+ can bind to the catalytic site? If only Fe2+ is selectively bound, then the Fe3+ conversion will not be a problem in the experiment because it is in the solution without binding to the protein.
Response 3: The immanent feature of the family 2-oxoglutarate and Fe(II) dependent dioxygenases is that they do require Fe2+ ions for their activity (Hausinger, RP Fe(II)/alpha-ketoglutarate-dependent hydroxylases and related enzymes (2004) Critical Reviews in Biochemistry and Molecular Biology 39: 21-68; DOI: 10.1080/10409230490440541). The problem in not in the presence of a trace amount of Fe3+ in the reaction mixtures, but in the uncontrolled depletion of Fe2+. However, this could be solved by addition of an antioxidant capable to Fe(III) reduction. We have used Fe2+/ascorbic acid whenever it was possible. So, there was no problem with Fe3+. Since the absorbance of ascorbic acid at 220-280 range nm is significant, no spectral detection of label-free protein was possible. So, nanoDSF and size exclusion chromatography experiments were carried on with Mn2+, which do not experience spontaneous oxidation and can be used without any antioxidant.
Point 4: Ca2+ affects FTO catalytic activity
Line61-67: “Although the stabilizing effect of Ca2+ was rather small, it affected regions of the N-terminal domain of BESFTO involved in enzymatic activity. Indeed, Ca2+ was found to decrease the activity of both FTO preparations (Figure 5A,B). Despite its marginal effect on FTO stability, Ca2+ decreased the demethylase activity significantly, in the case of BESFTO by more than 80%. Moreover, the stronger effect observed for BESFTO agreed with our in silico and biophysical predictions that phosphoserines 256 and 260 contribute to Ca2+ binding.
█ As mentioned above, the activity of ECFTO and BESFTO was decreased by Ca2+. Meanwhile, the author said, “our in silico and biophysical predictions that phosphoserines 256 and 260 contribute to Ca2+ binding. Insisted. There are no phosphosrines in ECFTO, how do I understand it?
Response 4: Our predictions are supported by HDX-MS data. Please note, that BESFTO is only partially phosphorylated at S256 and S260. However, calcium visibly stabilizes both ends of the loop containing both these serine residues (see Figure 2B). This clearly indicates that calcium cations do interact with this loop, even when serine residues are mostly non-phosphorylated. In this context, one can expect that serine phosphorylation modulate Ca binding rather than switches it on/off. Such interpretation is presented in lines 406-417 of the manuscript.
Point 5: I suggest experimenting with other metals as well to address these overall uncertain results.
Response 5: It seems interesting to study the effect of numerous cations on FTO properties. However, we must stress that we have initially predicted that calcium may affect FTO, and then verified this hypothesis experimentally. No other cations were predicted to interact with FTO in other regions than catalytic center of His-tag. Because of that we have focused on calcium.

Round 3
Reviewer 2 Report
The authors addressed the reviewers' concerns. After minor corrections, I recommend to accept this manuscript for publication.
minor
Line 472: "E. coli" should be italicized.
Line 474: "E. coli BL-21" should be "E. coli BL21".
Table 1:
- Authors should correct "137 NaCl, 2.7 KCl" in PRP1 and PRP2.
- Remove "with" from TSA2, TSA3, TSA4
- Remove "buffer" from MST3 and MST4